# Solubility Enhancement of Atrazine by Complexation with Cyclosophoraose Isolated from *Rhizobium leguminosarum* biovar *trifolii* TA-1

**DOI:** 10.3390/polym11030474

**Published:** 2019-03-12

**Authors:** Yohan Kim, Vijay Vilas Shinde, Daham Jeong, Youngjin Choi, Seunho Jung

**Affiliations:** 1Department of Systems Biotechnology & Dept. of Bioscience and Biotechnology, Microbial Carbohydrate Resource Bank (MCRB), Center for Biotechnology Research in UBITA (CBRU), Konkuk University, Seoul 05029, Korea; shsks1@hanmail.net (Y.K.); amir@konkuk.ac.kr (D.J.); 2Institute for Ubiquitous Information Technology and Applications (UBITA), Center for Biotechnology Research in UBITA (CBRU), Konkuk University, Seoul 05029, Korea; vijay.shinde040@gmail.com; 3Department of Food Science and Technology, Hoseo University, Asan 31499, Korea; ojchoi@hoseo.edu

**Keywords:** cyclosophoraose, atrazine, solubility, inclusion complex

## Abstract

*Rhizobium leguminosarum* biovar *trifolii* TA-1, a kind of soil bacteria, produces cyclosophoraoses (Cys). Cyclosophoraoses contain various ring sizes with degrees of polymerization ranging from 17 to 23. Atrazine is a hardly-soluble herbicide that contaminates soil and drinking water, and remains in soil for a long time. To remove this insoluble contaminant from aqueous solutions, we have enhanced the solubility of atrazine by complexation with Cys. The complex formation of Cys and atrazine was confirmed using ^1^H nuclear magnetic resonance (NMR), Fourier transform infrared (FT-IR) spectroscopy, differential scanning calorimetry (DSC), field emission scanning electron microscopy (FE-SEM), rotating frame nuclear overhauser spectroscopy (ROESY), and molecular modeling studies. The aqueous solubility of atrazine was enhanced 3.69-fold according to the added concentrations (20 mM) of Cys, compared to the 1.78-fold enhancements by β-cyclodextrin (β-CD). Cyclosophoraoses as an excellent solubility enhancer with long glucose chains that can effectively capture insoluble materials showed a potential application of microbial polysaccharides in the removal of hazardous hardly-soluble materials from aqueous solutions in the fields of biological and environmental industry.

## 1. Introduction

Cyclosophoraoses (Cys) are a class of unbranched cyclic oligosaccharides, isolated from *Rhizobium leguminosarum* biovar *trifolii* TA-1 (*R. trifolii* TA-1). Cyclosophoraoses are composed of β-(1,2)-D-glucans, with ring sizes ranging from 17 to 23 in their degree of polymerization (DP) [1,2]. They are predominantly synthesized in the periplasmic space and transported to the extracellular space. Cyclosophoraoses play a significant role in regulating the osmolarity in response to osmotic shock from circumstances, and root nodule formation during the initial stage of nitrogen fixation [3,4,5]. Due to the ability of Cys to form complexes with a variety of hydrophobic guest molecules, they have been investigated for host–guest complexation studies [6,7,8,9]. In addition, they can be utilized by modifying their functional groups such as butyryl, methyl, and carboxy-methyl groups. Recent studies have suggested the increased capacity complexation of these functionalized Cys with guest molecules [10,11].

Cyclodextrins (CDs: α-CD, β-CD, and γ-CD) are cyclic oligosaccharides, which have been reported to exhibit inclusion complexation behavior due to their three-dimensional toroidal shape made of six to eight α-1,4-linked glucose units [12]. The inclusion complexation ability of CDs improves the bioavailability of hardly-soluble compounds. In particular, β-cyclodextrin (β-CD) is commercially known to show not only enhanced inclusion complexing ability and water solubility, but also lower toxicity. β-CD is a cyclic heptasaccharide, containing α-1,4-linked glucose units, and is produced by bacterial enzymes using starch. β-CD has a unique property in that its inner cavity is hydrophobic, whereas the outer cavity is hydrophilic [13]. Based on this structural property, they are able to form inclusion complexes with hydrophobic compounds. This means that the hydrophobic inner cavity of β-CD interacts non-covalently with hydrophobic compounds [14]. Furthermore, the solubility and bioavailability of hydrophobic materials can be improved by forming inclusion complexes with β-CD [15,16]. Therefore, the solubility enhancement of CDs can be compared to that of Cys, which is a novel host molecule containing a higher number of glucose units for bulky guest molecules.

Atrazine (2-chloro-4-ethylamino-6-isopropylamino-1,3,5-triazine) is a representative *s*-triazine herbicide with good herbicidal ability available at a low price. For agricultural purposes, it is widely used to control weeds, particularly for corn, sugarcane, and sorghum production [17,18,19]. Atrazine is one of the most common herbicides detected in soil and drinking water worldwide [20,21] because it is sparingly soluble, and can remain in soil for a long time. In addition, the inappropriate use of herbicides is a common reason of food contamination [22]. Therefore, atrazine was banned in Europe in 2004 because it was frequently found in soil, surface waters, subterranean waters, and food; however, it is still used in the United States [23]. As atrazine, which is an endocrine disruptor, can disturb the natural hormonal system [24], the correlation between prenatal atrazine exposure and increased risk of cardiovascular disease and diabetes have been reported in infants [25]. The other effects of atrazine have been reported to include the induction of cell apoptosis and neurodegenerative diseases [26]. It also affects various living organisms by binding to the growth hormone receptor [27]. It also showed a potential for genotoxicity in the goldfish *Carassius auratus* [28]. Some studies indicated a relation between atrazine and a reduction in rat testicular sperm production and cell viability [29], in addition to myocardial angiogenesis [30]. Therefore, many studies focused on investigating techniques to reduce the toxicity of atrazine in several ways.

In order to remove this contaminant from soil and water, many extracting techniques have been proposed. Because most pollutants are sparingly soluble, the methanol (ethanol) water co-solvent system and surfactant Triton X-100 are commonly used and show a high removal efficiency [31,32,33]. However, using organic solvents and surfactants have the disadvantage of their toxicity to humans and the environment. Hence, the water rinsing method that uses a solubility enhancer is considered the best way to remove these pollutants in an environment-friendly way [34,35]. As atrazine is also practically insoluble, there are some studies that have focused on increasing the solubility of atrazine using normal CDs and CDs anchored on silica, forming a hybrid material [36,37,38]. Therefore, application of CDs for water treatment have been reported in several review articles due to the relatively hydrophobic environment of their interior cavity [39]. Recently, environmental applications of CD polymer were suggested for wastewater treatment [40,41]. It has been previously reported that Cys have a higher potential to form an inclusion complex with hardly-soluble guest molecules than CDs. Because Cys contain longer glucose chains, their flexibility has an effect on high water solubility. In this regard, this research could set precedent on research on the use Cys polymers for water treatment.

In this study, we applied the remarkable solubility enhancer Cys to atrazine and compared its performance with that of CDs. We plan to proceed with the removal of the atrazine using various derivative polymers of Cys in further studies.

## 2. Materials and Methods

### 2.1. Materials

Atrazine and CDs were purchased from Sigma–Aldrich Chemicals Co. (St. Louis, MO, USA). Dimethyl sulfoxide-d_6_ (DMSO-d_6_, 99.9 atom % D) was purchased from Cambridge Isotope Laboratories, Inc. (Andover, MA, USA). All other chemicals were of reagent grade and used without further purification.

### 2.2. Preparation of Cyclosophoraoses (Cys)

Cyclosophoraoses were obtained from the Microbial Carbohydrate Resource Bank (MCRB) at Konkuk University, Korea. The isolation and purification of Cys from *R. trifolii* TA-1 were conducted as described in previous studies [42,43]. *Rhizobium trifolii* TA-1 was grown in 1 L of GMS (glutamate, mannitol, and salts) medium with 50 g mannitol and 10 g glutamic acid at 25 °C for 14 days. Cells were harvested using centrifugation at 8000× *g* for 15 min and concentrated culture supernatants up to 5-fold using rotary evaporation, respectively. To remove high-molecular-weight (HMW) glycans, the concentrated culture supernatants were precipitated by adding three volumes of ethanol. Furthermore, the HMW glycans were separated from the concentrated sample using centrifugation. The supernatant was concentrated up to 10-fold using rotary evaporation. Cyclosophoraoses in the concentrated sample was precipitated by adding 10 volumes of ethanol. The precipitate was dissolved in distilled water and chromatographed on a Bio-Gel P-6. The fractions containing Cys were obtained and assayed using the phenol–sulfuric acid method. Cyclosophoraoses were desalted on a Bio-Gel P-2. Purified Cys were confirmed using the matrix-assisted laser desorption/ionization-time-of-flight (MALDI-TOF) mass spectrometry (Voyager-DETM STR Bio-Spectrometry, Applied Biosystems, Framingham, MA, USA) in the positive-ion mode using 2,5-dihydroxybenzoic acid (DHB) as the matrix. The information of polymerization degree of Cys was shown in Appendix A. The ^1^H nuclear magnetic resonance (NMR) spectroscopy (Bruker 500 MHz spectrometer, AMX, Karlsruhe, Germany) was used to confirm Cys in DMSO-d_6_ solvent.

### 2.3. ^1^H Nuclear Magnetic Resonance (NMR) Spectroscopy

The NMR spectroscopic analysis was carried out on a Bruker 500 MHz spectrometer (AMX, Karlsruhe, Germany) at 298 K. The purified Cys, atrazine, and atrazine/Cys complex were dissolved in DMSO-d_6_. The chemical shifts were calculated according to the formula: Δδ = δ(complex) − δ(free), where δ(free) and δ(complex) are the chemical shifts of atrazine in the absence and presence of Cys, respectively.

### 2.4. Phase Solubility Analysis

Due to the low water solubility of atrazine, it was dissolved in a solution (1 mL) of acetone: triply distilled water (TDW) in a ratio 4:1. To adjust different concentrations of Cys (0, 4, 8, 12, 16, and 20 mM), Cys were added to the atrazine solution (20 mM). The suspensions were magnetically stirred at 25 °C for 24 h, protected from light to prevent the decomposition of the molecules. After equilibrium was reached, evaporated acetone and the mixture was lyophilized. The lyophilized sample was dissolved in water and filtered using a polyvinylidene fluoride (PVDF) 0.2-mm filter (Whatman). Each sample was analyzed using a spectrophotometer (UV2450, Shimadzu Corporation, Kyoto, Japan) at a wavelength of 222 nm to measure the dissolved atrazine concentration. The graphs of concentrations of atrazine and Cys were plotted using the obtained data. The stability constant, $K_{c}$, for the complex formation was calculated from the linear portion of the solubility diagram using the Higuchi and Connors equation (Equation (1)) [44]. This method was used again to study the thermodynamics of the phase solubility study.
(1)$$K_{c}=\frac{\mathrm{Slope}}{S_{0}\left(1-\mathrm{Slope}\right)}$$
Δ*G* (kJ/mol), Δ*H* (kJ/mol), and Δ*S* (kJ/mol·K) were calculated depending on temperature using Equation (2).

Δ*G*° = − R*T* lnK = Δ*H*° − *T* Δ*S*°(2)

### 2.5. Job’s Plot Analysis 

The stoichiometry of the complex was measured using the continuous variation Job’s plot method. The sum of the concentrations of both components was maintained constant ([Cys] + [atrazine] = 10 mM) and the molar fraction (*r* = [Cys]/[Cys] + [atrazine]) was varied from 0.0 to 1.0. The stoichiometric ratio was obtained by plotting △*Abs* ⋯ *r* (where △Abs is the difference in the atrazine absorbance with and without Cys), and by finding the *r* value corresponding to this dependence [45,46].

### 2.6. Fourier Transform Infrared (FT-IR) Spectroscopy 

The Fourier transform infrared (FT-IR) spectra were obtained using a Bruker IFS-66/S spectrometer (Bruker, Karlsruhe, Germany) with potassium bromide (KBr) pellets as support in the scanning range of 650–4000 cm^−1^.

### 2.7. Differential Scanning Calorimetry (DSC)

The thermal behavior of atrazine, Cys, and atrazine/Cys complex was then examined using a DSC 7020 (SEICO INST., Chiba, Japan). A sample of 5 mg was placed into sealed aluminum pans prior to heating under nitrogen (40 mL min^−1^) at a scanning rate of 10 °C min^−1^. The observations were recorded over the temperature range of 20 °C to 220 °C. 

### 2.8. X-ray Diffraction Analysis (XRD)

X-ray diffraction analysis (XRD) was performed on a Bruker D8 DISCOVER diffractometer (Bruker, Karlsruhe, Germany) while using Cu–Kα radiation. It recorded XRD patterns by analyzing diffractions at a 2θ angle values between 10° and 30° in 1° min^−1^ increments and a recording time.

### 2.9. Field Emission Scanning Electron Microscopy (FE-SEM)

A Hitachi S-4700, produced by the Hitachi High-Technologies Corporation (Tokyo, Japan), was used to carry out the field emission scanning electron microscopy (FE-SEM). To fix the samples on a brass stub, a double-sided adhesive carbon tape was used. The powder samples were coated on the surface of a thin gold layer. The images were photographed at an excitation voltage of 10 kV.

### 2.10. Rotating Frame Nuclear Overhauser Spectroscopy (ROESY)

The 2D ^1^H—^1^H ROESY spectrum of the inclusion complex of atrazine with Cys was recorded using 256/2048 complex data points and a pulse train to attain a spin-lock field with a mixing time of 400 ms for the complex. The NMR analyses were carried out on the Bruker Avance 600 MHz spectrometer (AMX, Karlsruhe, Germany) in DMSO-d_6_ solvent at 25 °C.

### 2.11. Molecular Modeling

The structure of Cys was constructed using the MacroModel product in Schrodinger suite 2018-1. The initial monomeric Cys (DP = 19) structure was edited using the 3D-Builder module of Maestro 11.5.011. The molecular structure of the Cys monomer was energy-minimized and subjected to the conformational searching module in the MacroModel. An appropriate model for Cys was developed with OPLS2005 forcefield under the mixed torsional/large-scale low-frequency mode by maximum 1000 iterations. The flexible-ligand docking simulation of atrazine upon Cys was performed using the Glide product in the Schrodinger software [47]. The molecular structure of atrazine was edited using the 3D-Builder module of the Maestro program. The Receptor Grid Generation tool was used to generate a grid box covering the Cys structure. The Glide docking score was obtained using the extra XP mode function suitable for accurate pose prediction. In this docking job, 0.5 kcal/mol of energy window and distance-dependent dielectric constant (ε = 1) was applied to the docked pose sampling.

## 3. Results

### 3.1. Characterization of Atrazine and Cys

After cultivation, separation, and purification as described in the experimental methods section, Cys were obtained and their structures were analyzed using the NMR spectroscopy. In the ^1^H—^1^H correlation spectroscopy (COSY) and ^1^H–^13^C heteronuclear single quantum correlation (HSQC) spectra, the chemical structures of Cys were confirmed by clear correlation (Appendix A). In the NMR spectrum, the assignments of the proton signals of atrazine, Cys, and atrazine/Cys inclusion complex are shown in Figure 1. Figure 1b shows the characteristic peaks of Cys by designating each proton on glucose unit [48]. Whole peaks of the atrazine/Cys inclusion complex are shown in Figure 1c, and can be compared to spectra shown in Figure 1a,b. In the NMR spectrum of atrazine (Figure 1a), a- and b-NH protons appear in the range of 7.00–8.00 ppm, and c-H and d-H protons are visible at 4.006 ppm and 3.229 ppm, respectively. The terminal protons of e, f-H are visible at 1.113 ppm and 1.086 ppm, respectively. The NMR spectrum data of Cys in DMSO-d_6_ are shown in Figure 1b [8]. In the atrazine/Cys inclusion complex, shown in Figure 1c, the whole peaks correspond to the peaks of atrazine and Cys. In particular, a- and b-NH protons can be exactly designated in the range of 7.00–8.00 ppm. These phenomena of overlapping and duplicated peak patterns can be explained by four distinct conformational isomers of a-NH (ethylamino proton) and b-NH (isopropylamino proton) [49]. The peaks of each remaining solvent appeared at approximately 3.337 ppm (water), 2.502 ppm (DMSO-d_6_), and 2.093 ppm (acetone). In addition, chemical shifts of atrazine by Cys upon the inclusion complexation were measured for the inclusion complexation study (Appendix A). These characteristic peaks suggested that the inclusion complex was formed due to the wrapping of Cys around atrazine at a specific location.

### 3.2. Phase Solubility Tests

Phase solubility diagrams of atrazine with Cys, α-CD, β-CD, and γ-CD were obtained using the UV spectra of different sample concentrations (Figure 2). The plot of atrazine and Cys appeared to be A_L_ type, showing direct proportionality between the concentration of Cys and solubility of atrazine. The linear graph derived from atrazine and Cys suggests a 1:1 molecular association. The solubilizing effect of Cys was approximately 1.78 times that of β-CD. The respective stability constants, K, were determined using Equation (1) (Appendix A). Δ*G* (kJ/mol), Δ*H* (kJ/mol), and Δ*S* (kJ/mol·K) were determined using Equation (2), depending on temperature (Appendix A). These values are summarized in Appendix A. The effect of temperature on complex stability was studied by measuring phase solubility diagrams at different temperatures (298.15, 303.15, 308.15, and 313.15 K). It was observed that complex formation with Cys was largely driven by favorable enthalpy change (ΔH = −2.0027 kJ/mol) and entropy change (ΔS = 0.0372 kJ/mol). As temperature increases, Gibbs free energy (ΔG) have the tendency to increase in the (-) value. This indicates that the inclusion complex of Cys with atrazine was thermodynamically favorable. The stoichiometry of the complex was assessed using the Job’s method. It can be seen from Figure 3 that the highest molar fraction was observed at 0.5, indicating that Cys and atrazine formed a complex in a 1:1 ratio, which was similar to the ratio obtained using the phase solubility diagram.

### 3.3. Fourier Transform Infrared (FT-IR) Spectroscopic Analysis

Fourier transform infrared spectroscopy was used to characterize the structure of atrazine/Cys and compared it to that of atrazine and Cys (Figure 4). The chemical interaction between the two molecules shows a distinguishable change in intensity, shape, and peak shift in the infrared spectrum. The IR spectrum of Cys shows the presence of a peak at 3400 cm^−1^, which is assigned to the O–H stretching vibration, and two peaks observed at 1650 cm^−1^ and 1074 cm^−1^ corresponding to the O–H bending and C–O stretching vibration, respectively. The IR spectra of atrazine containing 3252 cm^−1^, 2971 cm^−1^, 1541 cm^−1^, and 1167 cm^−1^ of N–H, C–H, triazine group, and C–N bonds, respectively, are listed in Table 1. The physical mixture of atrazine/Cys was used for comparison with the atrazine/Cys inclusion complex. The FT-IR spectrum of the physical mixture shows absorption peaks with a reduced intensity compared to the pure atrazine. Furthermore, Cys peaks also appeared at the same positions. This could be due to the simple addition of atrazine and Cys. However, in the atrazine/Cys inclusion complex, the characteristic peaks of atrazine disappeared because of the inclusion complex with Cys. This indicated that the physical properties of atrazine were modified due to the molecular interactions within the inclusion complex. This confirmed the complex formation of atrazine and Cys.

### 3.4. Differential Scanning Calorimetry (DSC) Analysis

The DSC analysis was used to characterize the interaction between the host and guest molecules in a solid state. Figure 5 shows the DSC thermograms of atrazine, Cys, atrazine/Cys physical mixture, and the atrazine/Cys inclusion complex. Atrazine shows a sharp endothermic peak at 176.3 °C, which is its melting point. Cyclosophoraoses show an endothermic broad peak between 130 and 150 °C. Both endothermic peaks of atrazine and Cys were also detected in the DSC curve of the atrazine/Cys physical mixture. The change in the form of melting peak on the DSC curve of the physical mixture can be explained by the solid–solid [50,51]. Therefore, the change in form of melting peak on the DSC curve of the physical mixture might be affected by Cys as a solid state. However, the endothermic peak of atrazine disappears in the DSC curve of the atrazine/Cys inclusion complex. The vanishing whole peak of atrazine might be due to the forming inclusion complex completely [52,53]. These results suggest that the formation of complexes containing atrazine and Cys induced a change in the crystal state of atrazine.

### 3.5. X-ray Diffraction (XRD) Analysis 

The influence of the crystalline behavior of atrazine and Cys were studied by XRD. Figure 6 shows the XRD diagrams of atrazine, Cys, atrazine/Cys physical mixture, and atrazine/Cys inclusion complex. The characteristic diffraction peaks of atrazine were appeared at 2*θ* around 12.2°, 18.1°, 19.4°, 22.7°, 23.9°, and 27.1°. Also, the characteristic peaks of Cys appeared at 2*θ* around 10.4°. In the atrazine/Cys physical mixture, the intensity of these characteristic peaks decreased, but still appeared. Some characteristic peaks of the atrazine disappeared around 12.2°, 22.7°, 23.9°, and 27.1° after the inclusion complex, but the shape appearance of the Cys was maintained. This indicates that new solid crystalline phases were created that correspond to an inclusion complex. Then, the XRD corroborated the results that were obtained from FT-IR, DSC, and the following FE-SEM experiment. These results imply the differences in the morphological surface characteristics before and after the inclusion complex.

### 3.6. Field Emission Scanning Electron Microscopy (FE-SEM) Analysis

The FE-SEM characterizes the morphology changes of the inclusion complexes. The SEM images of atrazine, Cys, atrazine/Cys physical mixture, and atrazine/Cys inclusion complex are shown in Figure 7. A crystalline-particle shape and a rough-plate shape can be observed on atrazine (Figure 7a) and Cys (Figure 7b), respectively. The atrazine/Cys physical mixture shows a mixed shape that seems like the particles of atrazine are on the rough plate of Cys (Figure 7c). However, the atrazine/Cys inclusion complex clearly shows a thin-plate-shaped surface morphology (Figure 7d), unlike atrazine and Cys. The differences in the flatness and morphology of the material can be observed in the images before and after the inclusion complex of atrazine and Cys. The degree of surface roughness can be recognized clearly from the FE-SEM data. The atrazine/Cys inclusion complex has a feature that appears to be a flat surface on a larger section. This type of change may be due to the complexation of atrazine using Cys. This result confirms the formation of atrazine/Cys inclusion complexes.

### 3.7. ROESY Spectroscopy of Atrazine/Cys Inclusion Complexes

The rotating frame nuclear Overhauser effect spectroscopy (ROESY) experiment was carried out to explain the intermolecular interaction within the inclusion complex [54]. The ROESY spectra of the complexes between atrazine and Cys are shown in Figure 8. In the ROESY experiments, the intermolecular cross peaks between the proton (5-H) of the Cys and the b-NH proton (isopropylamino proton) of atrazine represent the correlation by complexation. In Figure 8, clear cross-peaks are observed between the b-NH proton of atrazine at 7.706, 7.613, and 7.386 ppm, respectively, and the 5-H protons of Cys are observed at 3.226 ppm. In particular, the repetitive signals of b-NH are caused by four distinct atrazine conformational isomers that have side-chains of a-NH (ethylamino proton) and b-NH [49]. These results suggest that the b-NH proton of atrazine can be specifically interacted with the 5-H of Cys. These interactions imply the intermolecular interaction of the atrazine/Cys inclusion complex.

### 3.8. Molecular Modeling of Atrazine/Cys Inclusion Complexes 

Figure 9 shows the representative image for the binding pose of the atrazine/Cys complex. Because of the small size of atrazine, this compound was located in the molecular cavity derived from the cyclic feature of Cys. By complexation, the molecular surface area was reduced by 2.99 nm^2^, which means stable complex formation. Particularly, b-NH and Cl atoms of atrazine were facing inward of the Cys cavity. However, an a-NH atom of the atrazine was exposed to the outside while facing out of the Cys cavity. It can be seen that five glucopyranose rings of Cys surround atrazine and maintain a stable complex (Figure 9a,c). Generally, 5-C and 5-H in five rings are in close contact with atrazine. These molecular features correlated well with the NMR-derived experimental results [54].

## 4. Conclusions

In this study, an enhanced solubility of atrazine was achieved by forming an inclusion complex with Cys, which were produced from a kind of soil bacteria, *R. trifolii* TA-1, to remove atrazine from aqueous solutions. The stability constant of the atrazine/Cys inclusion complex was determined to be 197.648 M^−1^ using the phase solubility diagram. The Job’s plot method was used to determine the stoichiometry of the atrazine/Cys inclusion complex to be 1:1. The correlation between the protons of atrazine and the proton of Cys was assessed using ROESY analysis. The results indicated that Cys wrapped around atrazine in the complex, and molecular modeling studies supported this phenomenon. In addition, the atrazine/Cys inclusion complex was confirmed by using NMR, FT-IR spectroscopy, DSC, and FE-SEM. The aqueous solubility of atrazine was increased 3.69-fold up to 20 mM concentration of Cys, compared to the 1.78-fold increment by β-CD. 

These results suggested the possibility of further biological and environmental applications of Cys containing a large number of glucose residues isolated from rhizobial species to remove hardly-soluble hazardous materials from aqueous solutions. 

## Figures and Tables

**Figure 1 polymers-11-00474-f001:**
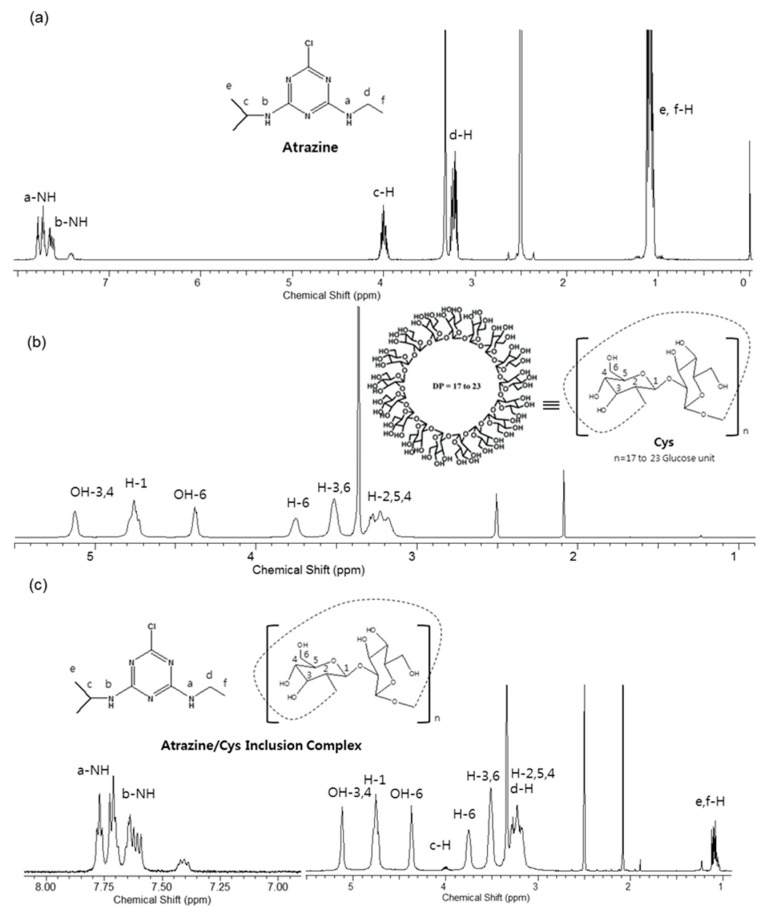
^1^H NMR spectra of (**a**) atrazine; (**b**) cyclosophoraoses (Cys); and (**c**) atrazine/Cys inclusion complex.

**Figure 2 polymers-11-00474-f002:**
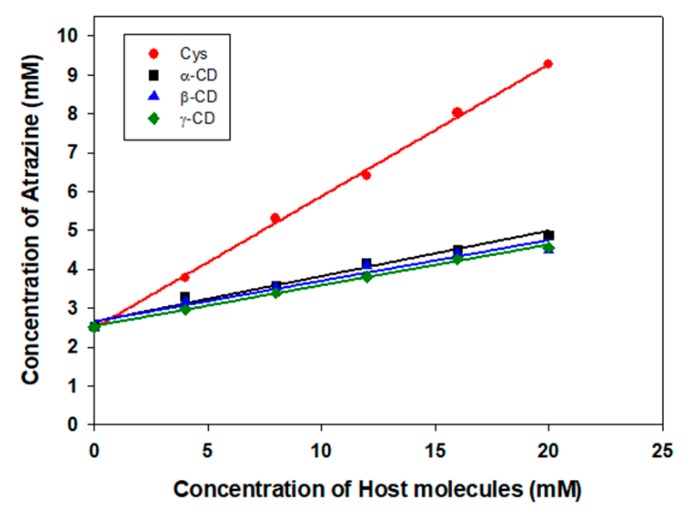
Phase solubility diagrams of atrazine in aqueous solution with Cys, α-CD, β-CD, and γ-CD at 25 °C.

**Figure 3 polymers-11-00474-f003:**
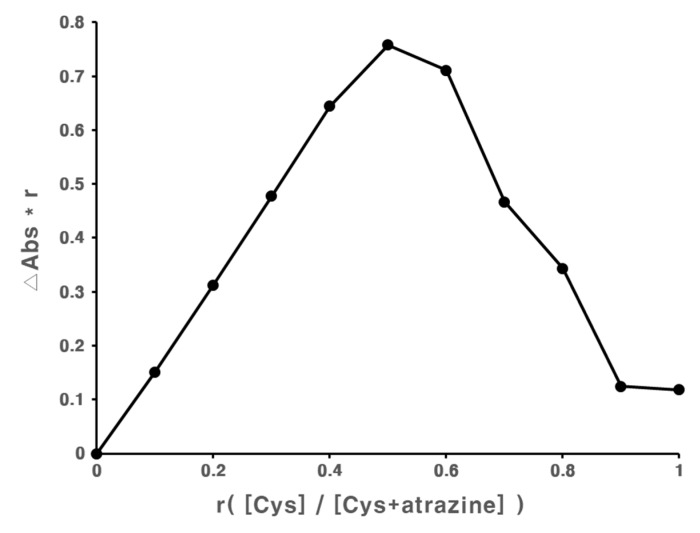
Job’s plot for atrazine and Cys.

**Figure 4 polymers-11-00474-f004:**
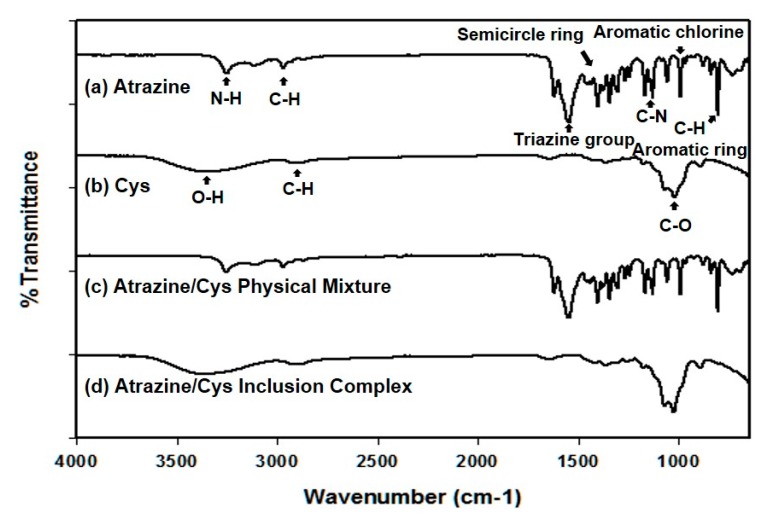
Fourier transform infrared (FT-IR) spectra of (**a**) atrazine; (**b**) Cys; (**c**) atrazine/Cys physical mixture; and (**d**) atrazine/Cys inclusion complex.

**Figure 5 polymers-11-00474-f005:**
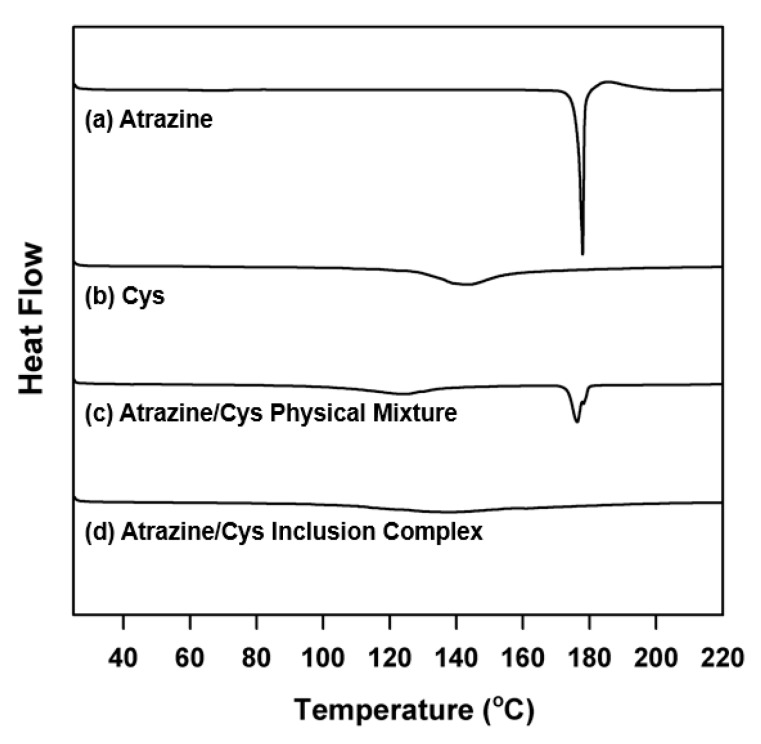
DSC curves of (**a**) atrazine; (**b**) Cys; (**c**) atrazine/Cys physical mixture; (**d**) atrazine/Cys inclusion complex.

**Figure 6 polymers-11-00474-f006:**
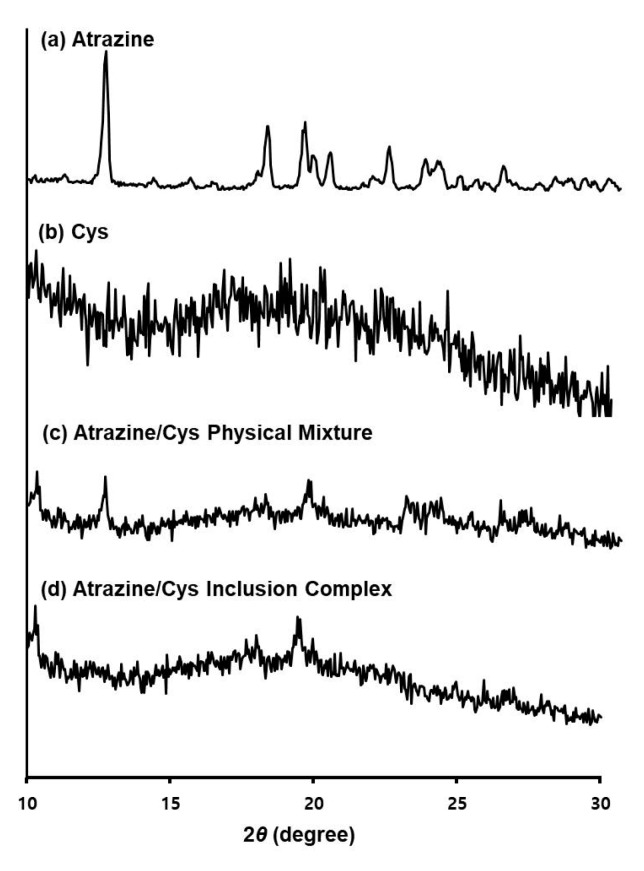
XRD analysis of (**a**) atrazine; (**b**) Cys; (**c**) atrazine/Cys physical mixture; (**d**) atrazine/Cys inclusion complex.

**Figure 7 polymers-11-00474-f007:**
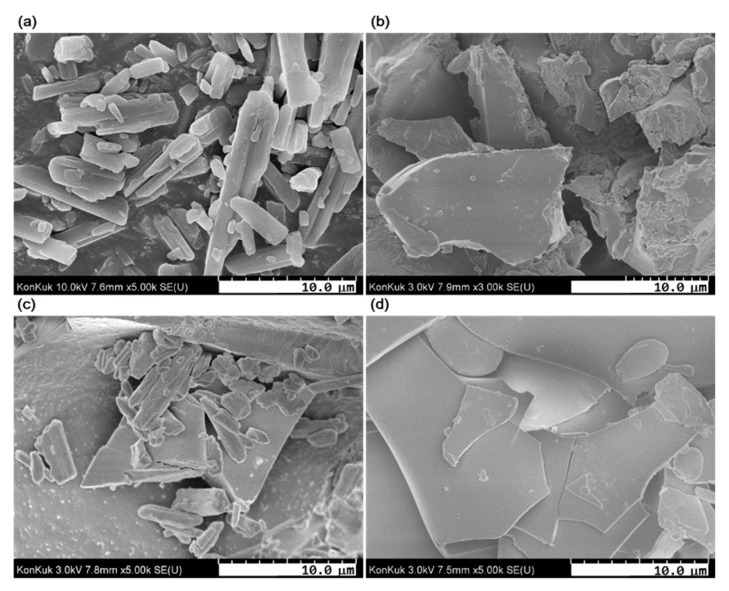
SEM images of (**a**) atrazine; (**b**) Cys; (**c**) atrazine/Cys physical mixture; (**d**) atrazine/Cys inclusion complex.

**Figure 8 polymers-11-00474-f008:**
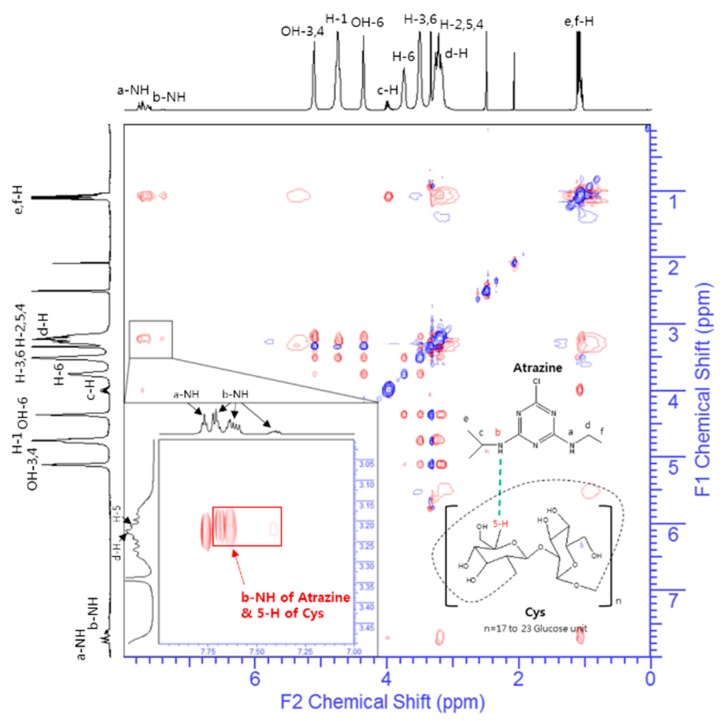
ROESY spectrum for atrazine/Cys inclusion complex.

**Figure 9 polymers-11-00474-f009:**
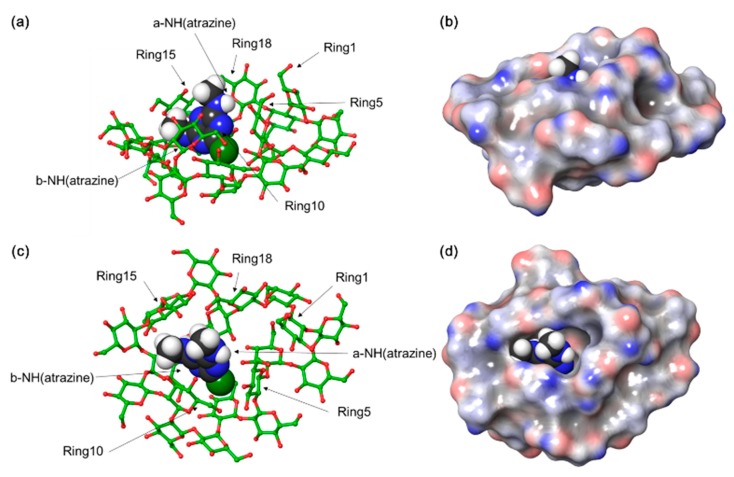
The highest-scoring docked pose of Cys and atrazine from molecular docking simulation: (**a**) side-view; (**b**) side-view with molecular surface area of Cys; (**c**) top-view; (**d**) top-view with molecular surface area of Cys. The carbon atoms of Cys were colored green, and atrazine was rendered by gray. Atrazine was represented by a space-filling model. For clarity, all hydrogen atoms of Cys were omitted.

**Table 1 polymers-11-00474-t001:** FT-IR absorption bands for atrazine, Cys, atrazine/Cys physical mixture, and atrazine/Cys inclusion complex.

Assignment	Atrazine (cm^−1^)	Cys (cm^−1^)	Atrazine/Cys Physical Mixture (cm^−1^)	Atrazine/Cys Inclusion Complex (cm^−1^)
N–H	3252		3251	
C–H	2971	2885	2843	2888
Triazine group	1541		1545	
Semicircle ring	1401		1412	
C–N	1167		1176	
Aromatic chlorine	1056		1058	
Triazine ring sextant	804		800	
C–H stretching of aromatic rings	730		729	
O–H stretching, bending		3200–3600, 1618		3200–3700, 1626
C–O		1056		1068

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
