# Peer review of "Solubility Enhancement of Atrazine by Complexation with Cyclosophoraose Isolated from Rhizobium leguminosarum biovar trifolii TA-1"

_polymers, 2019, doi:10.3390/polym11030474_

Round 1
Reviewer 1 Report
Manuscript title: Polymers-446797
Title: Solubility Enhancement of Atrazine by Complexation with Cyclosophoraose isolated from Rhizobium 3 leguminosarum biovar trifolii TA-1
Authors: Yohan Kim, Vijay Vilas Shinde, Daham Jeong, Youngjin Choi, and Seunho Jung 1
This manuscript is well planned and carried out to address one of the major challenges associated with waste water treatment.
Based on the following reasons, I recommend major revisions until the manuscript is improved for next round of consideration.
1. This reviewer is of the view that this manuscript is best suited for journals that report inclusion complexes (IC). Another aspect is that there is no polymer involved and is being submitted to a polymer journal.
2. In addition, this reviewer is not convinced with the analysis of some data. For example, the authors report that upon IC formation, the peak that is ascribed to atrazine vanishes. Typically, in an IC (or atleast with cyclodextrins), the peak intensity decreases, but not necessarily vanish. Also, typically there is a shift observed compared to neat host and physical mixture. At least provide citation justifying your analyses.
3. SEM should be accompanied by say HR-TEM (and SAED) analyses to show absence of crystallinity. Alternatively, XRD can also be utilized to show amorphousness of the IC (provided the stoichiometry is 1).
4. Please provide a justification how this can be CyS can be fabricated into a suitable device for water treatment.
Comments to the Author
Unfortunately, the manuscript needs to be revised extensively for typographical and syntax errors.
Author Response
Manuscripts no. polymer-446797
Response to Referees' Comments
Reviewer 1:
[1] This reviewer is of the view that this manuscript is best suited for journals that report inclusion complexes (IC). Another aspect is that there is no polymer involved and is being submitted to a polymer journal.
☞ As you comment, this article is about inclusion complex. However, the cys studied in this paper is a cyclic polysaccharide consisting of 17 to 23 glucose. In the recent issues of the ‘Polymers’ journal, there are publications about the inclusion complex of beta-cyclodextrin (seven glucose unit) similar to Cys, (Polymers 2019, 11(1), 145; Polymers 2019, 11(1), 71; Polymers 2018, 10(12), 1324; Polymers 2018, 10(12), 1294) so we thought it was right to submit our paper to this journal.
[2] In addition, this reviewer is not convinced with the analysis of some data. For example, the authors report that upon IC formation, the peak that is ascribed to atrazine vanishes. Typically, in an IC (or at least with cyclodextrins), the peak intensity decreases, but not necessarily vanish. Also, typically there is a shift observed compared to neat host and physical mixture. At least provide citation justifying your analyses.
☞ As your comment, we have added the following citations to make the data convincing. In previous studies, some papers have reported DSC data where guest molecule peak of IC formation became to be vanished after the complexation1,2. Especially, guest molecule peak was totally vanished when host molecule concentration (b-CD) was increased to 1:2 mol ratio (Fig. 5, KS1:2M) for the inclusion complexation. It might be due to the formation of the complete inclusion complex in the case of 1:2 mol ratio2 . Therefore, vanishing of guest molecule peak in DSC was a predictable and convincing result based on those two papers. Cys actually showed a higher ability to make an inclusion complex with the guest molecule (atrazine) than b-CD did.
Bertacche, Vittorio, et al. "Host–guest interaction study of resveratrol with natural and modified cyclodextrins." Journal of inclusion phenomena and macrocyclic chemistry 55.3-4 (2006): 279-287.
Nalluri, Buchi N., et al. "Physicochemical characterization and dissolution properties of nimesulide-cyclodextrin binary systems." AAPs PharmSciTech 4.1 (2003): 6.
BEFORE>
3.4. Differential Scanning Calorimetry (DSC) Analysis
The DSC analysis is used to characterize the interaction between the host and guest molecules in a solid state. Figure 5 shows the DSC thermograms of Atrazine, Cys, Atrazine/Cys physical mixture, and Atrazine/Cys inclusion complex. Atrazine shows a sharp endothermic peak at 176.3 °C, which is its melting point. Cys shows an endothermic broad peak between 130 °C and 150 °C. Both endothermic peaks of atrazine and Cys are also detected in the DSC curve of Atrazine/Cys physical mixture. However, the endothermic peak of atrazine disappears in the DSC curve of Atrazine/Cys inclusion complex. These results suggested that the formation of complexes containing atrazine and Cys induced a change in the crystal state of atrazine.
AFTER>
3.4. Differential Scanning Calorimetry (DSC) Analysis
The DSC analysis is used to characterize the interaction between the host and guest molecules in a solid state. Figure 5 shows the DSC thermograms of atrazine, Cys, atrazine/Cys physical mixture, and atrazine/Cys inclusion complex. Atrazine shows a sharp endothermic peak at 176.3 °C, which is its melting point. Cys show an endothermic broad peak between 130 and 150 °C. Both endothermic peaks of atrazine and Cys are also detected in the DSC curve of atrazine/Cys physical mixture. The change in the form of melting peak on the DSC curve of physical mixture can be explained by the solid–solid [49,50]. Therefore, the change in form of melting peak on DSC curve of physical mixture might be affected by Cys as solid state. However, the endothermic peak of atrazine disappears in the DSC curve of atrazine/Cys inclusion complex. Vanishing whole peak of atrazine might be due to forming inclusion complex completely [51,52]. These results suggested that the formation of complexes containing atrazine and Cys induced a change in the crystal state of atrazine.
[3] SEM should be accompanied by say HR-TEM (and SAED) analyses to show absence of crystallinity. Alternatively, XRD can also be utilized to show amorphousness of the IC (provided the stoichiometry is 1).
☞ As your comment, we performed and added the newly XRD experiment to support FE-SEM data in the revised manuscript.
AFTER>
3.5. X-ray Diffraction (XRD) Analysis
The influence of the crystalline behavior of atrazine and Cys were studied by XRD. Figure 6 shows the XRD diagrams of atrazine, Cys, atrazine/Cys physical mixture, and atrazine/Cys inclusion complex. The characteristic diffraction peaks of atrazine were appeared at 2θ around 12.2°, 18.1°, 19.4°, 22.7°, 23.9°, and 27.1°. Also, the characteristic peaks of Cys were appeared at 2 θ around 10.4°. In atrazine/Cys physical mixture, the intensity of these characteristic peaks were decreased, but still appeared. Some characteristic peaks of the atrazine were disappeared around 12.2°, 22.7°, 23.9°, and 27.1° after inclusion complex, but the shape appearance of the Cys was maintained. This indicates that the new solid crystalline phases were created that correspond to an inclusion complex. Then, the XRD corroborated the results that were obtained from FT-IR, DSC, and following experiment FE-SEM. These results imply the differences of morphological surface characteristics before and after inclusion complex.
(Please see the attached Word file.) Figure 6. XRD analysis of (a) atrazine; (b) Cys; (c) atrazine/Cys physical mixture; (d) atrazine/Cys inclusion complex.
[4] Please provide a justification how this can be CyS can be fabricated into a suitable device for water treatment.
☞ As your comment, we added the comments on the potential to use Cys as a suitable device for water treatment in the section of Introduction with the following citations in the revised manuscript.
Crini, Grégorio. "Recent developments in polysaccharide-based materials used as adsorbents in wastewater treatment." Progress in polymer science 30.1 (2005): 38-70.
Morin-Crini, Nadia, and Gregorio Crini. "Environmental applications of water-insoluble β-cyclodextrin–epichlorohydrin polymers." Progress in Polymer Science 38.2 (2013): 344-368.
Alsbaiee, Alaaeddin, et al. "Rapid removal of organic micropollutants from water by a porous β-cyclodextrin polymer." Nature 529.7585 (2016): 190.
BEFORE>
1. Introduction
In order to remove this contaminant from soil and water, many extracting techniques have been proposed. Because most pollutants are sparingly soluble, the methanol (ethanol) water co-solvent system and surfactant Triton X-100 were commonly used, and showed a high removal efficiency [31–33]. However, using organic solvents and surfactants have a disadvantage of their toxicity to humans and the environment. Hence, the water rinsing method that uses a solubility enhancer is considered the best way to remove these pollutants in an environment-friendly way [34,35]. As atrazine is also practically insoluble, there were some studies that focused on increasing the solubility of atrazine using normal cyclodextrins and anchored cyclodextrins on silica, forming a hybrid material [36–38]. However, this method is difficult to imply because of its less efficient performance. On the other hand, it has been previously reported that Cys has a good potential to form inclusion complex with hardly-soluble guest molecules. As Cys contain longer glucose chains than CDs, their flexibility have an effect on high water solubility and their further applications. In this study, we applied a remarkable solubility enhancer Cys to atrazine, and compared its performance with that of CDs (a-CD, b-CD, and g-CD). We will plan to proceed with the removal of the atrazine using various derivative polymers of Cys in further study.
AFTER>
1. Introduction
In order to remove this contaminant from soil and water, many extracting techniques have been proposed. Because most pollutants are sparingly soluble, the methanol (ethanol) water co-solvent system and surfactant Triton X-100 were commonly used, and showed a high removal efficiency [31–33]. However, using organic solvents and surfactants have a disadvantage of their toxicity to humans and the environment. Hence, the water rinsing method that uses a solubility enhancer is considered the best way to remove these pollutants in an environment-friendly way [34,35]. As atrazine is also practically insoluble, there were some studies that focused on increasing the solubility of atrazine using normal CDs and anchored CDs on silica, forming a hybrid material [36–38]. Therefore, applications of CDs for water treatment have been reported in several review articles due to relatively hydrophobic environment of interior cavity. [39]. Recently, environmental applications of CD polymer were suggested for wastewater treatment [40,41]. It has been previously reported that Cys have a higher potential to form inclusion complex with hardly-soluble guest molecules than CDs. Because Cys contain longer glucose chains, their flexibility have an effect on high water solubility. In this regard, this research could suggest a precedent research before Cys polymer for water treatment.
In this study, we applied a remarkable solubility enhancer Cys to atrazine, and compared its performance with that of CDs. We will plan to proceed with the removal of the atrazine using various derivative polymers of Cys in further study.
[5] Unfortunately, the manuscript needs to be revised extensively for typographical and syntax errors.
☞ As your comment, we extensively revised the typographical and syntax errors for abbreviation and capitalization principles for uniformity of context in the manuscript. These modified parts were easily found in the manuscript.
Thank you very much for your helpful comments.

Reviewer 2 Report
Manuscript prepared by Kim et al. is devoted to study on the interaction of atrazine with cyclosophoraose. Enhancement of atrazine solubility due to complexation with cyclosophoraose is considered. I think that major revision is necessary before the acceptance. My comments are below:
1. What is the polymerization degree of Cys?
2. The form of the melting peak on the DSC curve of physical mixture (c) was changed compared with the pure atrazine (a). What is the reason of this difference?
3. Phase solubility experiments were performed also with different CDs. Stability constants of the complexes with CDs should be calculated and analyzed.
4. Thermodynamic parameters of complex formation were calculated. However, they were not discussed.
5. There are some mistakes in the text:
line 129 - Higuchi and ConnorS equation
line 133 – The sentence is not correct “Using Equation 2, G (kJ/mol), H (kJ/mol), and S (kJ/mol*K) were calculated, depending on temperature.”
Line 213 – Caption to Fig. 2 is not complete (in what solvent? at what temperature?)
Author Response
Manuscripts no. polymer-446797
Response to Referees' Comments
Reviewer 2:
[1] What is the polymerization degree of Cys?
☞ As your comment, we provided the information of polymerization degree of Cys in the supplementary materials in the revised manuscript. Cys are composed of b-(1,2)-D-glucans, with ring sizes ranging from 17 to 23 in their degree of polymerization (DP).
(Please see the attached Word file.) Figure S1. MALDI-TOF mass spectra of Cys.
BEFORE>
2.2. Preparation of Cyclosohporaoses
Cys were obtained from the Microbial Carbohydrate Resource Bank (MCRB) at Konkuk University, Korea. The isolation and purification of Cys from Rhizobium leguminosarum biovar trifolii TA-1 were conducted as described in previous studies [39,40]. Rhizobium leguminosarum biovar trifolii TA-1 was grown in 1 L of GMS (glutamate, mannitol, and salts) medium with 50 g mannitol and 10 g glutamic acid at 25 °C for 14 days. Cells were harvested using centrifugation at 8000 × g for 15 min and and concentrated culture supernatants up to 5-fold using rotary evaporation, respectively. To remove high-molecular-weight (HMW) glycans, the concentrated culture supernatants were precipitated by adding three volumes of ethanol. Furthermore, the HMW glycans were separated from the concentrated sample using centrifugation. The supernatant was concentrated up to 10-fold using the rotary evaporation. Cys in the concentrated sample was precipitated by adding 10 volumes of ethanol. The precipitate was dissolved in distilled water and chromatographed on a Bio-Gel P-6. The fractions containing Cys were obtained and assayed using the phenol–sulfuric acid method. Cys were desalted on a Bio-Gel P-2. Purified Cys were confirmed using the MALDI-TOF (matrix-assisted laser desorption/ionization-time-of-flight) mass spectrometry (Voyager-DETM STR Bio-Spectrometry, Applied Biosystems, Framingham, MA, USA) in the positive ion mode using 2,5-dihydroxybenzoic acid (DHB) as the matrix. The NMR (nuclear magnetic resonance) spectroscopy (Bruker 500 MHz spectrometer, AMX, Germany) was used to confirm Cys in dimethyl sulfoxide (DMSO) solvent.
AFTER>
2.2. Preparation of Cyclosophoraoses (Cys)
Cys were obtained from the Microbial Carbohydrate Resource Bank (MCRB) at Konkuk University, Korea. The isolation and purification of Cys from R. trifolii TA-1 were conducted as described in previous studies [42,43]. R. trifolii TA-1 was grown in 1 L of GMS (glutamate, mannitol, and salts) medium with 50 g mannitol and 10 g glutamic acid at 25 °C for 14 days. Cells were harvested using centrifugation at 8000 × g for 15 min and and concentrated culture supernatants up to 5-fold using rotary evaporation, respectively. To remove high-molecular-weight (HMW) glycans, the concentrated culture supernatants were precipitated by adding three volumes of ethanol. Furthermore, the HMW glycans were separated from the concentrated sample using centrifugation. The supernatant was concentrated up to 10-fold using the rotary evaporation. Cys in the concentrated sample was precipitated by adding 10 volumes of ethanol. The precipitate was dissolved in distilled water and chromatographed on a Bio-Gel P-6. The fractions containing Cys were obtained and assayed using the phenol–sulfuric acid method. Cys were desalted on a Bio-Gel P-2. Purified Cys were confirmed using the matrix-assisted laser desorption/ionization-time-of-flight (MALDI-TOF) mass spectrometry (Voyager-DETM STR Bio-Spectrometry, Applied Biosystems, Framingham, MA, USA) in the positive ion mode using 2,5-dihydroxybenzoic acid (DHB) as the matrix. The information of polymerization degree of Cys was shown in Figure S1. The 1H nuclear magnetic resonance (NMR) spectroscopy (Bruker 500 MHz spectrometer, AMX, Germany) was used to confirm Cys in DMSO-d6 solvent.
[2] The form of the melting peak on the DSC curve of physical mixture (c) was changed compared with the pure atrazine (a). What is the reason of this difference?
☞ As your comment, we provided the proper citations to explain change in form of the melting peak on the DSC curve of physical mixture in the revised manuscript. The reason of this difference was probably due to the formation of inclusion complex. We corrected the result parts of DSC experiments like the followings in the revised manuscript.
... The DSC analysis is used to characterize the interaction between the host and guest molecules in a solid state. Figure 5 shows the DSC thermograms of atrazine, Cys, atrazine/Cys physical mixture, and atrazine/Cys inclusion complex. Atrazine shows a sharp endothermic peak at 176.3 °C, which is its melting point. Cys show an endothermic broad peak between 130 and 150 °C. Both endothermic peaks of atrazine and Cys are also detected in the DSC curve of atrazine/Cys physical mixture. The change in the form of melting peak on the DSC curve of physical mixture can be explained by the solid–solid [49,50]. Therefore, the change in form of melting peak on DSC curve of physical mixture might be affected by Cys as solid state. However, the endothermic peak of atrazine disappears in the DSC curve of atrazine/Cys inclusion complex. Vanishing whole peak of atrazine might be due to forming inclusion complex completely [51,52]. These results suggested that the formation of complexes containing atrazine and Cys induced a change in the crystal state of atrazine.
Park, Kyeong Hui, et al. "Enhancement of Solubility and Bioavailability of Quercetin by Inclusion Complexation with the Cavity of Mono‐6‐deoxy‐6‐aminoethylamino‐β‐cyclodextrin." Bulletin of the Korean Chemical Society 38.8 (2017): 880-889.
Novoa, Gelsys Ananay Gonzalez, et al. "Physical solid-state properties and dissolution of sustained-release matrices of polyvinylacetate." European journal of pharmaceutics and biopharmaceutics 59.2 (2005): 343-350.
Bertacche, Vittorio, et al. "Host–guest interaction study of resveratrol with natural and modified cyclodextrins." Journal of inclusion phenomena and macrocyclic chemistry 55.3-4 (2006): 279-287.
Nalluri, Buchi N., et al. "Physicochemical characterization and dissolution properties of nimesulide-cyclodextrin binary systems." AAPs PharmSciTech 4.1 (2003): 6.
BEFORE>
3.4. Differential Scanning Calorimetry (DSC) Analysis
The DSC analysis is used to characterize the interaction between the host and guest molecules in a solid state. Figure 5 shows the DSC thermograms of Atrazine, Cys, Atrazine/Cys physical mixture, and Atrazine/Cys inclusion complex. Atrazine shows a sharp endothermic peak at 176.3 °C, which is its melting point. Cys shows an endothermic broad peak between 130 °C and 150 °C. Both endothermic peaks of atrazine and Cys are also detected in the DSC curve of Atrazine/Cys physical mixture. However, the endothermic peak of atrazine disappears in the DSC curve of Atrazine/Cys inclusion complex. These results suggested that the formation of complexes containing atrazine and Cys induced a change in the crystal state of atrazine.
AFTER>
3.4. Differential Scanning Calorimetry (DSC) Analysis
The DSC analysis is used to characterize the interaction between the host and guest molecules in a solid state. Figure 5 shows the DSC thermograms of atrazine, Cys, atrazine/Cys physical mixture, and atrazine/Cys inclusion complex. Atrazine shows a sharp endothermic peak at 176.3 °C, which is its melting point. Cys show an endothermic broad peak between 130 and 150 °C. Both endothermic peaks of atrazine and Cys are also detected in the DSC curve of atrazine/Cys physical mixture. The change in the form of melting peak on the DSC curve of physical mixture can be explained by the solid–solid [49,50]. Therefore, the change in form of melting peak on DSC curve of physical mixture might be affected by Cys as solid state. However, the endothermic peak of atrazine disappears in the DSC curve of atrazine/Cys inclusion complex. Vanishing whole peak of atrazine might be due to forming inclusion complex completely [51,52]. These results suggested that the formation of complexes containing atrazine and Cys induced a change in the crystal state of atrazine.
[3] Phase solubility experiments were performed also with different CDs. Stability constants of the complexes with CDs should be calculated and analyzed.
☞ As your comment, we provided information of stability constants of the complexes with CDs in the supplementary materials in the revised manuscript.
Table S2. Calculated data of phase solubility for Kc of Cys, a-CD, b-CD, and g-CD.
S0(M) | slope | Kc (M-1) | |
Atrazine/Cys | 2.4872 | 0.3394 | 206.5677 |
Atrazine/a-CD | 2.6458 | 0.1172 | 50.1774 |
Atrazine/b-Cys | 2.6497 | 0.1048 | 44.1819 |
Atrazine/g-Cys | 2.5353 | 0.1047 | 46.1263 |
BEFORE>
3.2. Phase Solubility Tests
Phase solubility diagrams of atrazine with Cys, a-CD, b-CD, and g-CD are obtained using the UV spectra of different sample concentrations (Figure 2). The plot of atrazine and Cys appears to be AL type, showing direct proportionality between the concentration of Cys and solubility of atrazine. The linear graph derived from atrazine and Cys suggests a 1:1 molecular association. The solubilizing effect of Cys was approximately 1.78 times that of b-CD. The respective stability constants, K, were determined using Equation (1). DG (kJ/mol), DH (kJ/mol), and DS (kJ/mol*K) were determined using Equation (2), depending on temperature (Figure S3). These values are summarized in Table S3. The stoichiometry of the complex was assessed using the Job’s method. It can be seen from Figure 3 that the highest molar fraction is observed at 0.5, indicating that Cys and atrazine formed a complex in 1:1 ratio, which was similar to the ratio obtained using the phase solubility diagram.
AFTER>
3.2. Phase Solubility Tests
Phase solubility diagrams of atrazine with Cys, a-CD, b-CD, and g-CD are obtained using the UV spectra of different sample concentrations (Figure 2). The plot of atrazine and Cys appears to be AL type, showing direct proportionality between the concentration of Cys and solubility of atrazine. The linear graph derived from atrazine and Cys suggests a 1:1 molecular association. The solubilizing effect of Cys was approximately 1.78 times that of b-CD. The respective stability constants, K, were determined using equation (1) (Table S2). DG (kJ/mol), DH (kJ/mol), and DS (kJ/mol*K) were determined using equation (2), depending on temperature (Figure S3). These values are summarized in Table S3. The effect of temperature on complex stability was studied by measuring phase solubility diagrams at different temperatures (298.15, 303.15, 308.15, and 313.15 K). It is observed that complex formation with Cys is largely driven by favorable enthalpy change (DH = -2.0027 kJ/mol) and entropy change (DS = 0.0372 kJ/mol). As temperature increase, Gibbs free energy (DG) have tendency to increase in the (-) value. This indicates that the inclusion complex of Cys with atrazine is thermodynamically favorable. The stoichiometry of the complex was assessed using the Job’s method. It can be seen from Figure 3 that the highest molar fraction is observed at 0.5, indicating that Cys and atrazine formed a complex in 1:1 ratio, which was similar to the ratio obtained using the phase solubility diagram.
[4] Thermodynamic parameters of complex formation were calculated. However, they were not discussed.
☞ As your comment, we provided discussion of thermodynamic parameters of complex formation in the revised manuscript.
BEFORE>
3.2. Phase Solubility Tests
Phase solubility diagrams of atrazine with Cys, a-CD, b-CD, and g-CD are obtained using the UV spectra of different sample concentrations (Figure 2). The plot of atrazine and Cys appears to be AL type, showing direct proportionality between the concentration of Cys and solubility of atrazine. The linear graph derived from atrazine and Cys suggests a 1:1 molecular association. The solubilizing effect of Cys was approximately 1.78 times that of b-CD. The respective stability constants, K, were determined using Equation (1). DG (kJ/mol), DH (kJ/mol), and DS (kJ/mol*K) were determined using Equation (2), depending on temperature (Figure S3). These values are summarized in Table S3. The stoichiometry of the complex was assessed using the Job’s method. It can be seen from Figure 3 that the highest molar fraction is observed at 0.5, indicating that Cys and atrazine formed a complex in 1:1 ratio, which was similar to the ratio obtained using the phase solubility diagram.
AFTER>
3.2. Phase Solubility Tests
Phase solubility diagrams of atrazine with Cys, a-CD, b-CD, and g-CD are obtained using the UV spectra of different sample concentrations (Figure 2). The plot of atrazine and Cys appears to be AL type, showing direct proportionality between the concentration of Cys and solubility of atrazine. The linear graph derived from atrazine and Cys suggests a 1:1 molecular association. The solubilizing effect of Cys was approximately 1.78 times that of b-CD. The respective stability constants, K, were determined using equation (1) (Table S2). DG (kJ/mol), DH (kJ/mol), and DS (kJ/mol*K) were determined using equation (2), depending on temperature (Figure S3). These values are summarized in Table S3. The effect of temperature on complex stability was studied by measuring phase solubility diagrams at different temperatures (298.15, 303.15, 308.15, and 313.15 K). It is observed that complex formation with Cys is largely driven by favorable enthalpy change (DH = -2.0027 kJ/mol) and entropy change (DS = 0.0372 kJ/mol). As temperature increase, Gibbs free energy (DG) have tendency to increase in the (-) value. This indicates that the inclusion complex of Cys with atrazine is thermodynamically favorable. The stoichiometry of the complex was assessed using the Job’s method. It can be seen from Figure 3 that the highest molar fraction is observed at 0.5, indicating that Cys and atrazine formed a complex in 1:1 ratio, which was similar to the ratio obtained using the phase solubility diagram.
[5] There are some mistakes in the text:
line 129 - Higuchi and ConnorS equation
line 133 – The sentence is not correct “Using Equation 2, G (kJ/mol), H (kJ/mol), and S (kJ/mol*K) were calculated, depending on temperature.”
Line 213 – Caption to Fig. 2 is not complete (in what solvent? at what temperature?)
☞ As your comment, we corrected some mistakes in the revised manuscript. Thank you.
BEFORE>
2.4. Phase Solubility Analysis
Due to the low water solubility of atrazine, it was dissolved in the solution (1 mL) of acetone : TDW (triply distilled water) in ratio 4:1. To adjust different concentrations of Cys (0, 4, 8, 12, 16, and 20 mM), Cys was added to the atrazine solution (20 mM). The suspensions were magnetically stirred at 25 °C for 24 h, protected from light to prevent the decomposition of the molecules. After equilibrium was reached, evaporated acetone and the mixture was lyophilized. The lyophilized sample was dissolved in water and filtered using a PVDF (polyvinylidene fluoride) 0.2 mm filter (Whatman). Each vial was analyzed using a spectrophotometer (UV2450, Shimadzu Corporation) at a wavelength of 222 nm to measure the dissolved atrazine concentration. The graphs of concentrations of atrazine and Cys were plotted using the obtained data. The stability constant, , for the complex formation was calculated from the linear portion of the solubility diagram using the Higuchi and Connor equation (Equation 1) [41]. This method was used again to study the thermodynamics of the phase solubility study.
(1)
Using Equation 2, DG (kJ/mol), DH (kJ/mol), and DS (kJ/mol*K) were calculated, depending on temperature.
DG° = − RT lnK = DH° - TDS° (2)
(Please see the attached Word file.)
Figure 2. Phase solubility diagrams of Atrazine with Cys, a-CD, b-CD, and g-CD.
AFTER>
2.4. Phase Solubility Analysis
Due to the low water solubility of atrazine, it was dissolved in the solution (1 mL) of acetone : triply distilled water (TDW) in ratio 4:1. To adjust different concentrations of Cys (0, 4, 8, 12, 16, and 20 mM), Cys were added to the atrazine solution (20 mM). The suspensions were magnetically stirred at 25 °C for 24 h, protected from light to prevent the decomposition of the molecules. After equilibrium was reached, evaporated acetone and the mixture was lyophilized. The lyophilized sample was dissolved in water and filtered using a polyvinylidene fluoride (PVDF) 0.2 mm filter (Whatman). Each samples was analyzed using a spectrophotometer (UV2450, Shimadzu Corporation) at a wavelength of 222 nm to measure the dissolved atrazine concentration. The graphs of concentrations of atrazine and Cys were plotted using the obtained data. The stability constant, , for the complex formation was calculated from the linear portion of the solubility diagram using the Higuchi and Connors equation (equation 1) [44]. This method was used again to study the thermodynamics of the phase solubility study.
(1)
DG (kJ/mol), DH (kJ/mol), and DS (kJ/mol*K) were calculated depending on temperature using equation 2.
DG° = − RT lnK = DH° - TDS° (2)
(Please see the attached Word file.) Figure 2. Phase solubility diagrams of Atrazine in aqueous solution with Cys, a-CD, b-CD, and g-CD at 25 °C.
Thank you very much for your helpful comments.

Round 2
Reviewer 2 Report
Manuscript can be accepted for publucation. For values reported in Table 2, no need to give a lot of numbers after the decimal point.
Author Response
Manuscripts no. polymer-446797
Response to Referees' Comments
Reviewer 2:
[1] Manuscript can be accepted for publication. For values reported in Table 2, no need to give a lot of numbers after the decimal point.
☞ As your comment, we revised the values reported in all the table including Table S2. We corrected all the values of tables in two after the decimal point in the revised manuscript.
BEFORE>
Table S2. Calculated data of phase solubility for Kc of Cys, a-CD, b-CD, and g-CD.
S0(M) | slope | Kc (M-1) | |
Atrazine/Cys | 2.4872 | 0.3394 | 206.5677 |
Atrazine/a-CD | 2.6458 | 0.1172 | 50.1774 |
Atrazine/b-Cys | 2.6497 | 0.1048 | 44.1819 |
Atrazine/g-Cys | 2.5353 | 0.1047 | 46.1263 |
Table S3. Calculated data of phase solubility for Kc, DG0, DH0, and DS0 at 298.15, 303.15, 308.15 and 313.15 K.
T (K) | S0(M) | slope | Kc (M-1) | DG0(kJ/mol) | DH0(kJ/mol) | DS0(kJ/mol*K) | |
Atrazine/Cys | 313.15 | 3.7569 | 0.4170 | 190.3863 | -13.6667 | -2.0027 | 0.0372 |
308.15 | 3.5573 | 0.4068 | 192.7774 | -13.4804 | |||
303.15 | 3.2668 | 0.3947 | 199.6033 | -13.3494 | |||
298.15 | 2.6712 | 0.3919 | 241.2647 | -13.5991 |
AFTER>
Table S2. Calculated data of phase solubility for Kc of Cys, a-CD, b-CD, and g-CD.
S0(M) | slope | Kc (M-1) | |
Atrazine/Cys | 2.48 | 0.33 | 206.56 |
Atrazine/a-CD | 2.64 | 0.11 | 50.17 |
Atrazine/b-Cys | 2.64 | 0.10 | 44.18 |
Atrazine/g-Cys | 2.53 | 0.10 | 46.12 |
Table S3. Calculated data of phase solubility for Kc, DG0, DH0, and DS0 at 298.15, 303.15, 308.15 and 313.15 K.
T (K) | S0(M) | slope | Kc (M-1) | DG0(kJ/mol) | DH0(kJ/mol) | DS0(kJ/mol*K) | |
Atrazine/Cys | 313.15 | 3.75 | 0.41 | 190.38 | -13.66 | -2.00 | 0.03 |
308.15 | 3.55 | 0.40 | 192.77 | -13.48 | |||
303.15 | 3.26 | 0.39 | 199.60 | -13.34 | |||
298.15 | 2.67 | 0.39 | 241.26 | -13.59 |
Thank you very much for your helpful comments.
